# A Solution for the Remote Care of Frail Elderly Individuals via Exergames

**DOI:** 10.3390/s21082719

**Published:** 2021-04-12

**Authors:** Marco Trombini, Federica Ferraro, Matteo Morando, Giovanni Regesta, Silvana Dellepiane

**Affiliations:** 1Electrical, Electronics and Telecommunication Engineering and Naval Architecture Department (DITEN), Università degli Studi di Genova, Via all’Opera Pia 11a, I-16145 Genoa, Italy; marco.trombini@edu.unige.it (M.T.); federica.ferraro@edu.unige.it (F.F.); morando.matteo@gmail.com (M.M.); 2Centro di Riabilitazione Gruppo Fides, Via Bolzano 1/C, I-16166 Genoa, Italy; giovanniregesta@gruppofides.it

**Keywords:** exergames, IoT solution, elder care, telerehabilitation

## Abstract

Internet of Things (IoT) solutions are a concrete answer to many needs in the healthcare framework since they enable remote support for patients and foster continuity of care. Currently, frail elderly people are among end users who most need and would benefit from IoT solutions from both a social and a healthcare point of view. Indeed, IoT technologies can provide a set of services to monitor the healthcare of the elderly or support them in order to reduce the risk of injuries, and preserve their motor and cognitive abilities. The main feature of IoT solutions for the elderly population is ease of use. Indeed, to fully exploit the potential of an IoT solution, patients should be able to autonomously deal with it. The remote-monitoring validation engineering system (ReMoVES) described here is an IoT solution that caters to the specific needs of frail elderly individuals. Its architecture was designed for use at rehabilitation centers and at patients’ homes. The system is user-friendly and comfortably usable by persons who are not familiar with technology. In addition, exergames enhance patient engagement in order to curb therapy abandonment. Along with the technical presentation of the solution, a real-life scenario application is described referring to sit-to-stand activity.

## 1. Introduction

The ongoing digital transformation in our society has significant impact on several technological aspects, such that the term Fourth Industrial Revolution has been used for a few years. As another revolution [1], Internet of Things (IoT) solutions are becoming increasingly relevant, and their use is consistently growing in several application domains. Regarding healthcare, the IoT market size was valued at USD 147.1 billion in 2018 [2], and is expected to reach USD 534.3 billion by 2025, expanding at a compound annual growth rate (CAGR) of 19.9% over the forecast period [3], and resulting in an estimated USD 63 billion of savings due to the deployment of medical IoT by 2022 [4]. All this is due to growing investments in digital technology implementation at healthcare institutions that address the need for the care of a growing geriatric population [5] coupled with the rising prevalence of chronic conditions [6]. In addition, the recent outbreak of COVID-19 has had strong impact on the health system, which had to adapt itself to various needs such as guaranteeing access to care for patients in forced quarantine or in solitary confinement, and meeting the needs for social distancing and reduction in access to healthcare facilities. Medical IoT solutions are an essential tool for responding to patient care needs under safe conditions. Hence, applications such as telemedicine, remote patient monitoring, and interactive medicine have a precise and crucial position in the fight against the coronavirus, such that several nations officially recommended their use [7].

The key benefit of the IoT in the medical domain is connected technology. Devices are used for assessing patients’ conditions, and monitoring and supporting rehabilitation, so that a personalized plan of care can be defined and kept updated. This also fosters continuity of care, enabling a patient to be supervised by a multidisciplinary team even after dehospitalization. The most ubiquitous of such devices are wearable or robotic devices, for instance, smart bands for data collection related to some physical activity [8] or other wearables for motion analysis, which can be devoted to specific body-part rehabilitation (e.g., shoulders [9] and knees [10]). Even though a deep interest in such devices is manifested in the healthcare context, wearables, robotic devices, and devices based on smartphone interaction are not very suitable for the elderly population or for dehospitalized and disabled patients. Indeed, to fully exploit the potential of an IoT solution, patients should be able to deal with it autonomously; however, the presence of wearable devices or controllers means that some external support may be needed for such activities. Thus, from the social and healthcare points of view, frail elderly individuals are among end users who most need and would benefit from easy-to-use IoT solutions. Indeed, frailty in the elderly corresponds to a broad clinical issue that concerns the physical, cognitive, and social aspects of the patient, particularly for people over the age of 75 [11]. The study by Fried et al. [12] defined a phenotype and thereby some characteristic traits of frailty in the elderly. Specifically, frailty is considered if at least 3 of the following symptoms are present: unintentional weight loss, fatigue, reduction in muscle strength, slower walking speed, and decreased physical activity. In cases where fewer than 3 of the symptoms are detected, one can speak of prefrailty. Frailty, therefore, differs from disability because it is characterized by a decline in several physiological aspects. Thus, in this sense, disability manifests itself more as a consequence of frailty itself. For this purpose, IoT technologies can provide a set of services to monitor elderly healthcare and behavior, and to reduce the risk of injuries. For instance, Tao et al. studied fall prediction based on human biomechanical equilibrium by analyzing data acquired by a Microsoft Kinect sensor installed in elderly individuals’ homes [13].

In such a framework, this paper presents the remote-monitoring validation engineering system (ReMoVES; numip.it/removes) [14], developed at the Università degli Studi di Genova. ReMoVES is a telerehabilitation platform that provides a set of IoT-based services to support motor and cognitive maintenance and recovery through exergames and digital versions of standard rehabilitation tests, carried out via Microsoft Kinect, Leap Motion, and a touchscreen. ReMoVES is based on a multiclient/-server architecture that allows for both the collection of and access to information from different locations. It was designed for use in rehabilitation centers with the help of clinical staff or even independently in the patient’s home, thus also enabling continuity of care after dehospitalization. In contrast to other solutions, ReMoVES is an auxiliary tool that provides therapists with objective data even when they cannot directly supervise their patients, such as during unattended use at their homes, along with automated data-processing techniques [15]. The role of exergames is important when referring to rehabilitative practice. Exergames are video games designed to promote physical activity, with users performing physical exercises [16]. They recently gained large popularity and proved to have scientific reliability, thus overcoming their original goal of mere entertainment. Furthermore, clinical evidence showed positive results for the preservation and improvement of cognitive functions in elderly populations [17]. In addition, gamification [18] proved to be effective for increasing engagement in activities in several domains [19]. As a result, it also fosters a motivating environment in order to keep patient engagement high without inducing boredom or fatigue, which may lead to frustration and the abandonment of therapy. ReMoVES is currently used in five centers (hospitals, clinics, and facilities), involving more than 200 patients and resulting in more than 2000 rehabilitation sessions. Various studies are being conducted that use ReMoVES, such as those concerning unilateral spatial neglect [20] and systemic sclerosis [21]. Its ease of use and technical reliability are supported by good emotional feedback provided by the patients who practiced ReMoVES exergames, and by the small number of registered technical failures (7 in 3 years).

The present study is contextualized in the framework of user-generated content (UGC) analysis. Therefore, it finds applications in research on the use of data science (DS) in digital marketing (DM). In 2020, nine topics for future research on DS in the DM ecosystem were defined [22]. Among them, four are in line with this work, i.e., medical-data eHealth; people: movement, organization, and personalization; the IoT; and new machine-learning model development.

The main contributions of this work are:developing and applying ReMoVES in the context of frail elderly individuals’ care, meeting the needs of the target population;demonstrating the ease of use of the presented solution, which allows for frail elderly people to autonomously access the provided services;proving that the remote monitoring of frail elderly individuals is possible with such a solution, thus fostering continuity of care;showing how off-the-shelf and inexpensive devices, such as the ones employed by ReMoVES, can be used for the satisfactory monitoring of patients;confirming that game-based activity enhances patient engagement, driving them to also practice the exercises unattended, including in elderly populations;showing how technology can foster socialization in elderly populations;reporting how ReMoVES simplifies the work of clinical specialists and promotes the establishment of individualized healthcare plans;describing a real-life scenario referring to the well-known sit-to-stand (STS) activity;defining the indicators used for the implicit analysis of the game session, which is involved in designing ad hoc data-processing techniques;analyzing the presented data from both qualitative and quantitative points of view with respect to elderly people and comparing them with results in the literature;releasing data from rehabilitation sessions via exergames, and inviting other researchers to leverage on the published database in order to establish the framework for research activities in such a context.

Following some preliminary applications such as the present one, ReMoVES is now being applied in several studies involving a large number of patients.

The manuscript is structured as follows. Section 2 briefly recaps the clinical motivation supporting the present work. In Section 3, other solutions are described, aimed at showing the technical background of IoT technologies for elderly care, and highlighting how ReMoVES and the present study differ from them. ReMoVES is described in Section 4 from a technical and an application point of view. The experimental phase is described in Section 5, focusing on analysis of the rehabilitation sessions of patients practicing the exergame for the STS. Similar works in the literature are compared to prove the reliability of the present study. Lastly, Section 6 discusses and concludes the present work.

## 2. Clinical Motivation

As the average age of the global population is growing, the global healthcare system has to respond to the need of elderly populations [23,24]. Indeed, due to age and related cognitive impairments, weakness is a major limiting factor related to daily life activities. For instance, the reduction in torque generation is reported at the level of the elbow, shoulder, fingers, and thumb, which worsens due to prolonged physical inactivity. Furthermore, simple activities, such as standing up, may be affected, causing falling risk and insecure gait. In addition to cases of psychiatric and neurological diseases, cognitive abilities inevitably decline in a healthy elderly population, thus leading to severe social and economic impact.

In this context, strength training associated with task-oriented training can intensify rehabilitation and reinforcement [25].

The study of Erickson et al. [26] suggested that physical exercise can produce cognitive improvements (associated with an increase in hippocampal volume) in accordance to [27] about increased levels of brain-derived neurotrophic factor (BDNF) in response to exercise.

By design, exergames are appropriate for this aim as they require the patient to produce physical movements in order to complete a task-oriented exercise in response to visual cues [28]. They are simultaneously able to improve patient engagement and train multiple cognitive processes [29].

## 3. Existing Solutions and Differences

The aim of the present section is to introduce information and communication technology (ICT)/IoT solutions in the context of frail elderly people’s care, highlighting major points in common and differences between those and ReMoVES.

The interest toward the well-being of the elderly is well-documented in the literature from environmental [30] and independent-living [31] perspectives. As health issues and frailty symptoms start arising, it is crucial to take action in effective ways.

Traditionally, a great portion of physical therapy, rehabilitation, and assessment is based on a clinician’s observations and judgment. Sensor and computing technologies that can be used for motion capture, performance assessment, and range-of-movement (ROM) measurements have drastically advanced in the past few years.

In such a context, many works focus on the early detection of frailty to help caregivers and patients, and address such a problem from several points of view. For instance, the authors of [32] evaluated the acceptability of solutions for detecting signs of frailty on the basis of techniques and clinical practice described in the literature. Their conclusion was that minimal clinic interruption, low requirements for resources, and added benefit (e.g., stratify risk, enhanced understanding of frailty) yield to higher acceptability. Such requirements are met by ReMoVES since it can be used in combination with other clinical needs, it is based on off-the-shelf components, and provides therapists with feedback that helps them in assessing patients’ conditions and defining further steps of the rehabilitation process.

The study in [33] highlighted the importance of frailty detection in a home-based context aimed at supporting the independent living of elderly individuals at home. The local-to-central unit architecture of ReMoVES and its ease of use were specifically designed to deliver home-based activities, thus facilitating dehospitalization and promoting continuity of care.

In [34], a platform for favoring personalized interventions to frail elderly persons was introduced. It deploys dashboards for doctors and patients, giving them control of the system and data visualization, respectively. Such a solution has some points in common with ReMoVES, with particular focus on its application layer that is described in Section 4.1.

Along with the spread of solutions targeting healthy and active aging, the need for aligning and combining the informative content obtained by different platforms arises. On the basis of [35], the information-uniformity issue was tackled in the fundamental study by Madueira et al. [36]. Indeed, the proposed My-AHA software is able to integrate multiple healthy and active aging platforms, thus providing a solution to be used in conjunction with other technologies. Similarly, ReMoVES was designed to be integrated with complementary devices, for instance, for biometric measurements or medical teleconsultations. In addition, ReMoVES directly encompasses activities in the form of exergames that can be practiced by patients.

There are several solutions in the context of exergames for clinical practice, for example, applications for poststroke [37,38,39,40] and Parkinson’s rehabilitation [41,42], and with particular emphasis on physical activity [43].

Despite the deep interest in such a topic, each solution is focused towards particular needs, developed independently from the others, and this makes it difficult to compare them. Furthermore, publications that provide raw data acquired during the execution of exergames were not found. The poor similarity of such solutions and their high variability, already pointed out in [36], along with the substantial lack of available data, limit the development of assistive computing technologies based on exergames, affecting both the technical aspect and the scientific innovation of the results. In particular, the possibility of comparing body-tracking methodologies (e.g., articulations to track; frequency), data-transfer solutions (e.g., client–server architecture or local storage only), and data-log formats (e.g., JSON or CSV) is hindered. Furthermore, the impossibility of accessing collected data in many studies proposed by other researchers in recent years precludes the use of big-data processing algorithms.

For the above reasons, the present paper highlights the potential of ReMoVES rather than comparing it with other solutions, in addition to describing its solid practical background. ReMoVES is used in several healthcare facilities, and doctors and therapists claim it is a very important tool, allowing for delivering more accurate measurements than what humans can collect without the need for uncomfortable and hard-to-use devices such as wearables and robotics.

The authors of the present manuscript aspire to sensitize the scientific community on the aforementioned issues, and invite other researchers to leverage the published database in order to establish a framework for research activities related to the use of exergames for rehabilitation, and to lay the groundwork for a direct comparison with systems similar to ReMoVES.

## 4. Materials and Methods

The present section goes over the materials and experimental setting of this manuscript. First, Section 4.1 introduces all of ReMoVES’ components and layers, describing how they were developed and their functionalities. Then, the setting of the experimental phase is presented in Section 4.2, along with study indicators that are of interest from a clinical point of view (Section 4.3). Such an application is only an example of the possible deployment of ReMoVES. All available activities are practiced by patients, but STS is the most comprehensive and investigated for clinical treatment.

### 4.1. ReMoVES Architecture

ReMoVES was developed by the Electrical, Electronics and Telecommunication Engineering and Naval Architecture Department (DITEN) of Università degli Studi di Genova [14]. The proposed IoT system provides a personalized rehabilitation program that can be performed at home by the patient, while the therapist can track training performance and effectiveness from any Internet-connected device. By developing game-based rehabilitation tools that are tailored to the therapy goals of different patient categories, multidimensional rehabilitation teams can be provided with more meaningful performance data. Among others, the monitoring of eventual compensation movements allows for the evaluation of whether an exercise is correctly performed.

Several IoT architectures for telemedicine systems and e-health were proposed in the literature [44,45,46], but the most compliant with ReMoVES is the one composed of four layers shown in Figure 1.

These levels work closely together, and ensure the archiving, processing, monitoring, and proper evaluation of patients’ rehabilitation performance. The four-layer architecture divides the connection part from the server/cloud part. It is important to define the correct role of the latter because the physical server used in ReMoVES plays a fundamental role in the correct processing and management of the entire IoT system.

A detailed description of the four layers follows referring to the used technologies and devices.

#### 4.1.1. Sensor Layer

The bottom layer is the sensor or perception layer and consists of the patient client. It deals with the management of so-called “things” (i.e., sensors connected to the system). ReMoVES employs off-the-shelf devices, i.e., Microsoft Kinect V2, Leap Motion, and a touchscreen, resulting in a low-cost solution for telerehabilitation. These devices are installed and connected to a computer, and through simple body gestures or touches (in the case of touchscreen), the patient interacts with the game shown on the screen. Patient movements are recorded without requiring the intrusive use of video cameras, which require specialized methods for tracking the user’s body, and are heavy and errorprone. After the patient finishes the game session, raw information is generated from tracked data and sent to the upper level. A brief description of the included sensors in the platform is provided. The real-scenario application here refers to full-body activity; so, exergames delivered via the Kinect sensors are described, furnishing particular details about the game used for the performance assessment of frail elderly people.

Microsoft Kinect V2: A motion-sensing input device based on a time-of-flight camera to build a depth map of the environment. It can simultaneously track in 3D up to 25 fundamental joints (Figure 2) of the framed human body. It offers a wide field of view (70${}^{\circ }$× 60${}^{\circ }$) and recognition up to 4.5 m from the device [47]. Data from the tracked user’s body are recorded at a frequency of 10 Hz. Several studies demonstrated that the Microsoft Kinect V2 can validly obtain spatiotemporal parameters [48,49]. Microsoft Kinect is also a satisfactory tool for rehabilitation due to its low cost and adequate spatial accuracy (with an order of magnitude of centimeters) [50].

Exergames based on Microsoft Kinect have a significant field of application in assistive technologies for the elderly, such as in reducing fall risk, improving physical performance, and reversing the deterioration process in frail and prefrail elderly persons [51].

Leap Motion: Explicitly aimed at the recognition of hand gestures, it calculates the position of the fingertips and the orientation of the hand. Its deployment in ReMoVES is devoted to hand-district rehabilitation exergames.

Touchscreen: Required for interacting with the subset of exergames for cognitive assessment. The monitor is positioned on a table with an angle to the plane of a few degrees. Cognitive exergames in the ReMoVES platform are a digital reinterpretation of some gold-standard tests administered on paper to patients. Interaction through a touchscreen monitor allows for complete data collection, also helping the administrator avoid taking notes during patient activity. Examples of collected auxiliary data are interaction speed and methods or strategies used by the patient to complete the test.

Exergames: The digital games were developed from scratch for this research. They encourage the patient to autonomously carry out functional exercises along with traditional motion rehabilitation. The creation of these activities involved different processes, technologies, and specialists. It is necessary to pay particular attention to the specifications provided by physiotherapists and physiatrists, who share their skills to define the requirements and parameters of the game. The present exergames are considered to be assessment and rehabilitation activities, delivering task-oriented training by requiring the patients to fulfil consecutive and repetitive tasks. They foster mild-intensity activity, which promotes active aging for frail elderly individuals, and allows for the preservation or reacquisition of functional skills that are involved in real-life activities.

To design an enjoyable and safe gaming experience for elderly users, several age-related requirements needed to be considered [52]: (i) The target audience’s lack of previous gaming experience: devices such as Microsoft Kinect enable users to control and naturally interact with exergames without the need to physically touch a game controller or object of any kind. Microsoft Kinect achieves this through a natural user interface by tracking the user’s body movements. (ii) Exergames should focus on a simple interaction mechanism, while complex and decorative graphics should be kept to a minimum. (iii) Exergames should avoid frustration and foster an enjoyable player experience: when the motor skills of the user are reduced, a preventive calibration phase allows for the patient to complete the game task even with a limited ROM.

The system currently includes six different exergames for the Kinect sensor that can be modified on the basis of level parameters, duration, range of motion, speed, or others. These activities automatically adapt to the patient capabilities due to a calibration phase. The thumbnails of Kinect exergames are shown in Figure 3.

For the sake of completeness, a brief description of the Kinect exergames follows.

Equilibrium Paint: this game is an interactive version of the sit-to-stand exercise. The user repeatedly stands up and sits down within a predefined amount of time (30 s). The scene shows a horizontal wooden beam on which paint cans are placed. The inclination of the beam directly depends on the angle of the patient’s shoulders during the STS, traced by Microsoft Kinect. When the patient does not symmetrically stand up, the paint cans fall down, causing a score penalty in the game.Owl Nest: the patient is encouraged to reach an on-screen target with an arm motion (reaching task) in order to achieve a high ingame score. Many colorful owls randomly appear in a position in the screen, and the user carries them to the nest to gain points. Then, other ones appear on the screen.Shelf Cans: introduces the patient to a virtual environment that is similar to a kitchen. With an arm movement, the patient grabs one of the colorful drink cans appearing in the middle of the screen and drags it to the corresponding shelf. This game is appealing because it requires the user to be attentive to drop off the drink can on the correct shelf according to its color.Hot Air: this is an activity to train the patient’s body balance. The user can control the direction of a hot-air balloon floating in the sky with the balance shift: ingame scores are collected when it is led towards the bonus targets.Push Box: assesses balance, where the patient must stretch forward with their arms parallel to the ground. It takes inspiration from a phase of the Berg balance test. The purpose of this exercise is to push a box into a hole a few meters in front of the box.Flappy Cloud: this is a functional exercise for the lower limbs. The leg abduction–adduction movement reflects the position of a cloud object in the game screen: the patient makes it move forward without hitting some obstacles.

#### 4.1.2. Network Layer

The role of the network layer is to establish communication between data tracked by the sensors and stored in a local PC and the remote server or cloud. In ReMoVES, data-log files in JavaScript Object Notation (JSON) format are temporarily stored in the local unit or PC installed in the patient’s home or at hospital. These data are sent to the central server as soon as an Internet connection is available via Ethernet or Wi-Fi. This functionality was added in order to combat any possible connection trouble and to facilitate domestic use where a reliable Internet connection may not be available.

#### 4.1.3. Server Layer

The server layer provides data elaboration and analysis via cloud or server storage. Software running on the physical ReMoVES server can manage content-independent data flow to be compliant with software reuse logic. Server software consists of a traditional Linux–Apache–MySQL–PHP (LAMP) stack, and provides data-storage solutions, data-processing methods, and a web application for clinicians to view information through dedicated graphic interfaces. The ReMoVES server has only three types of application programming interfaces (APIs) for the management of client or server data synchronization. Data communication is in secure mode based on hypertext transfer protocol secure (HTTPS). In HTTPS, the communication protocol is encrypted using transport layer security (TLS). Certificates are issued by the Let’s Encrypt authority. To process acquired information, a complete postmovement reconstruction of ingame events is allowed. Additionally, this component runs the data-processing algorithms and provides the interface for displaying the results.

Database: This subsection describes the MySQL relational database used by ReMoVES. The dataset consists of a structured collection of JSON files, each of them containing an array of temporal events. In each element of the array, there are key-value pairs that provide data. Some keys are common to all exergames, such as time of recording in milliseconds (ms), ingame score, and Kinect joint position (see Figure 2). In addition, other keys are provided depending on the game.

#### 4.1.4. Application Layer

This layer consist of the therapist client, which provides therapists, physiotherapists, and doctors with direct access to data. Specifically, the built-in algorithms provide a clear and concise report to the therapist in order to facilitate the interpretation of therapy evolution. The web interface provides a user-friendly means for the clinical staff to consult information, also displaying patient performance in graphic mode, and to assign rehabilitation therapies. The layout dynamically adapts to the size and type of device; this allows for connection even from a smartphone in the case that the therapist does not have an available computer.

Figure 4 and Figure 5 show pictures of hardware and software architectures, respectively. The patient client is composed of a local unit with the following hardware requirements:processor, seventh generation Intel^®^ Core™i5 CPU (quad-core 2.4 GHz or faster);memory, 4 GB RAM;storage, 20 GB;video card, DirectX11-capable from NVIDIA, AMD, or Intel with at least 1 GB VRAM; anddedicated USB3 port.

Microsoft Kinect or Leap Motion sensors are connected to the computer on the basis of the therapist’s recovery plan; thereby, exergames are assigned to the patient. A touchscreen monitor is added instead in the case of assessment through cognitive tests. As mentioned in Section 4.1.2, an Internet connection is not mandatory for the user identification phase and to locally start the exergames, but it is necessary for data synchronization with the server. The central unit is composed of the ReMoVES server, which performs the data synchronization with the patient or client units, and stores and processes data in the MySQL database. Therapists can access from any device the web application supplied by WEB server functions.

### 4.2. Experiment Setting

In the following section, a real-world application of the system is provided for the condition assessment of elderly people referring to STS activity. The experiment setting is described here.

STS is a well-known assessment test of which the importance in estimating lower-limb strength is widely recognized [53]. Many studies discussed its effectiveness as an indicator of leg weakness and related falling risk in elderly and disabled people [54]. Indeed, it is included in the Berg balance test battery as a standard activity for frail elderly persons [55]. In addition to assessment, STS is a task-oriented and strength-reinforcing exercise [56].

For frail elderly people, such an exercise represents a simple and daily activity, but at the same time, it involves the muscles of the lower limbs, stimulating them and thus allowing for their strengthening over time. The importance of this test gave rise to numerous studies that implement new technologies, such as wearable sensors or baropodometer boards [57,58].

There exist two different STS protocols: the 30 s STS [59], consisting of standing and sitting as many times as possible; and the 5 times STS [60], requiring five complete STS cycles to be performed in the shortest time possible. Here, the 30 s STS is considered, and sessions by a population of frail elderly individuals referred to the Centro di Riabilitazione, Gruppo Fides Genova (Italy) are described. An ad hoc exergame, Equilibrium Paint, was developed in the context of the improved STS with the support of new technologies. It guides the patient to correctly conduct the exercise and improves engagement due to stimuli and visual feedback. Section 4.1.1 describes the exergame, and Figure 6 shows a captured screenshot of the game during activity.

Intervention using the ReMoVES exergame platform did not replace the classical rehabilitation program but is an integrative tool of usual treatment. Patient participation in the activity is uniquely aimed at collecting movement data during the execution of the STS exergame rather than undergoing clinical tests. Data were collected over a period lasting up to 2 months and up to twice a week, and admission to each game session was determined on the basis of the judgment of the physiatrist, who assessed the willingness to participate and the general conditions of the patient at that particular time.

### 4.3. Indicators

Some peculiar features are defined and extracted from the patients’ game sessions in order to provide a picture of the general conditions of the considered population. The definition of the considered features was inspired by works in the literature such as [61,62]. They were computed from the spatial coordinates of joints in Figure 2, where *x*, *y*, and *z* represent the mediolateral, anteroposterior, and vertical directions, respectively.

The main indicator is the number of sit-up occurrences (NSU) during the 30 s duration of the test. This is computed by analyzing the trajectory of the spine middle joint (joint 7 in Figure 2) along the vertical axis. Each peak of such a trajectory represents a sit-up. For the population under analysis, the average NSU is $\bar{NSU}=4.5$.

Peak detection also allows for separating the ascending and descending phases during activities. They are identified as the parts of the trajectory between a local minimum of the spine middle height and the following peak, and between a peak and the following local minimum of the spine middle height, respectively. In this fashion, it is possible to in-depth analyze both phases by computing ad hoc indicators.

The first feature that is introduced is the upper-body flexion angle (UBFA), which represents the angle of flexion of the trunk and is computed as
(1)$$UBFA=\arctan\frac{z_{2}-z_{7}}{y_{2}-y_{7}}.$$

The UBFA is maximal when the player is in a standing position, and reaches values of approximately 90${}^{\circ }$ when sitting. In addition, other values are present that represent the intermediate phase from a sitting to a standing position and vice versa. For standing up, the player should move forward, which results in a decrease in sitting UBFA.

Similarly to the UBFA, the indicator of the lower-limb flexion angle (LLFA) represents the knee angle, and can be computed for both the left and the right limb. It is defined as
(2)$$LLFA=180+\theta_{femur}-\theta_{tibia}.$$
where $\theta_{femur}=\arctan\frac{z_{20}-z_{19}}{y_{20}-y_{19}}$, $\theta_{tibia}=\arctan\frac{z_{22}-z_{20}}{y_{22}-y_{20}}$ for the left limb and $\theta_{femur}=\arctan\frac{z_{21}-z_{17}}{y_{21}-y_{17}}$, $\theta_{tibia}=\arctan\frac{z_{24}-z_{21}}{y_{24}-y_{21}}$ for the right limb. Variation in this angle for both the left and the right limb is similar to the trajectory of the spine middle joint (see Section 5.2).

During this activity, patients may adopt erroneous behavior such as moving the shoulders or hips. Hence, it is important that therapists supervising the rehabilitation are informed about these compensatory movements. Regarding shoulder movement, the upper-body twist angle (UBTA) depicts the angle of the line joining the shoulders (joints 3–5 in Figure 2) on the axial plane:(3)$$UBTA=\arctan\frac{y_{5}-y_{3}}{x_{5}-x_{3}}.$$

Hip displacement is calculated on the basis of the anteroposterior and mediolateral displacement of the center of mass (COM). The COM is defined as the middle point between the right and left hips (joints 17 and 19 in Figure 2, respectively) and spine middle (joint 7 in Figure 2):(4)$$\begin{array}{c} COM=\left(\bar{x},\bar{y},\bar{z}\right)=\left(\frac{x_{17}+x_{19}+x_{7}}{3},\frac{y_{17}+y_{19}+y_{7}}{3},\frac{z_{17}+z_{19}+z_{7}}{3}\right). \end{array}$$

Hence, indicators center-of-mass anteroposterior movement (COM AP) and center of mass mediolateral movement (COM ML) depict COM positions on the axial plane.

To conclude, upper-frame velocity (UfV) is the velocity of motion in either the ascending or descending phase. For one ascending phase, it is computed as
(5)$$UfV_{up}=\frac{z_{peak}-z_{localmin}}{time_{peak}-time_{localmin}}.$$

Similarly, in the descending phase, it is
(6)$$UfV_{down}=-\frac{z_{localmin}-z_{peak}}{time_{localmin}-time_{peak}}.$$

More generally, all aforementioned features are separately computed in the ascending and descending phases in order to provide a fragmented and specific analysis of the patients’ sessions.

## 5. Experiment Results

The aforementioned real-world case study is described in the present section. The current study involved 13 frail elderly people (6 females and 7 males) with an average age of 82.3±6.2 who participated several times to the rehabilitation sessions via ReMoVES. This was a preliminary feasibility study to evaluate the possible use of ReMoVES in a real-world scenario. Feedback from the present work drives further applications involving more patients. Participants reported that they felt safe while playing the game, and there were no adverse events while playing. Most of the patients stated that they enjoyed this extra activity, asking the clinical staff to participate more frequently. An interesting social interaction developed among the participants, who enjoyed watching others carry out the activities.

In addition, the collected data are released with this publication. They are available for download on GitLab (https://gitlab.com/NumIP/removes-fe-data/ accessed on 1 April 2021). Data are licensed under the Creative Commons Attribution 4.0 International license (CC BY-NC-SA 4.0). The data release is because, despite the deep research interest in this field, publications that provide raw data acquired during the execution of exergames were not found.

### 5.1. Implicit Activity Analysis

The implicit analysis of the activity performed by the involved patients is presented. Mean values of the proposed indicators were collected, and their coherence with already published results was statistically tested. In addition, aimed at enabling deeper analysis of each game session, a graphic visualization of the indicators along the time dimension is shown. Such graphs are provided to therapists via the application layer, so that clinical staff analyze both summary statistical indicators and patient performance during the whole session. In this fashion, even some erroneous movements or loss of energy, which may be limited to a short period of time, can be noted by the medical specialists, leading to a complete and deep clinical picture of the patients.

The average features of the available population are summarized in Table 1. Negative values for the UBTA indicate that the left shoulder was put forward while practicing the activity.

To address the coherence of the derived data with respect to the literature, the results of [61,62] were considered for the discussion. In [61], the indicators standing and sitting COM AP, standing and sitting COM ML, $UfV_{up}$, and $UfV_{down}$ were calculated with respect to a population of healthy elderly individuals (mean values were 0.01, 0.03, 0.03, and 0.04 cm, and 0.78 and 0.71 m/s, respectively). Hence, a statistical test was performed to verify the assumption that the indicator values in [61] depicted a better general health condition than the ones deduced for the population under analysis. A one-tailed t-test was used, and the assumption was confirmed with *p* value <0.01.

In addition, the authors in [62] showed mean values for the range of UBFA in both the ascending and descending phases in a population of frail elderly persons. Via a two-tailed t-test, the assumption that the mean ranges of UBFA in [62] and in the present work were equal was verified with *p* value <0.01.

### 5.2. Graphs on Therapist Client

As anticipated, therapists were also provided with graphs depicting all game sessions, delivering more comprehensive informative content than that in the mean or range indicators. An example of the graphic representation available on the therapist client is shown in Figure 7. In particular, Figure 7a depicts the trajectory of the COM and peaks; hence, corresponding standing positions were visible. In particular, parts with a light-gray background are for the ascending phase, and parts with a dark-gray background represent the descending phase. Figure 7b shows UBFA values during the session. Figure 7c shows LLFA values for both the right and the left limb during the session. The trend of this chart is very similar to that of the COM. A standing position also requires limbs to be fully extended, corresponding to the peaks of the LLFA indicators. Figure 7d presents the shoulder twist on the axial plane during the session. Lastly, COM AP and COM ML displacements are depicted in Figure 8 on the transverse plane. Reduced lateral displacement in the second graph with respect to the first suggests that the patient stabilized themselves while playing.

### 5.3. Analysis of Presented Graphs

The present section provides an interpretation of the graphs in Section 5.2. This allows for discussing the considered indicators and for highlighting how the present IoT solution can be used for remote monitoring.

The COM trajectory in Figure 7a shows that the patient performed a smooth movement with no particular pauses. The resulting regular path means that the patient did not experience particular fatigue and managed to control their motion. So, by only considering such a graph, a therapist would say that the patient’s performance was fairly good. However, Figure 7b,c, for UBFA and LLFA, respectively, depict incomplete movement. Indeed, the patient is supposed to reach maximal extension while standing, namely, the maximal values of UBFA and LLFA (corresponding to COM peaks) should reach approximately 180${}^{\circ }$. While LLFA satisfies such a requirement, meaning correct leg extension, the maximal values of UBFA were around 130${}^{\circ }$, denoting that the patient remained bent forward when standing. Figure 7d for UBTA depicts that shoulder rotations were very small, denoting correct movement (the patient is required to preserve shoulders in the frontal plane, i.e., without trunk rotations). To conclude, graphs in Figure 8 depicting AP and ML movements show that the patient was not laterally significantly displaced (about 2 cm), confirming the correct execution of the exercise apart from the vertical trunk extension.

This shows how multidimensional data can provide the clinical staff with precise information. This is very important for reliable remote monitoring, by which small or partially erroneous behaviors can also be detected and corrected.

## 6. Discussion and Conclusions

The present section is for the discussion and conclusions. Style and structure are inspired by [63].

IoT system ReMoVES administers exergames for motor and cognitive activities; its use in the context of frail elderly individuals’ care was presented here. By using low-cost off-the-shelf components and an easy-to-use interface, patient activity can be monitored even when executed without therapist supervision. Additionally, people who are not familiar with new technology can perform this activity according to their personalized plan of care. All this also allows for continuity of care after dehospitalization, and the remote supervision of patient activity by clinical staff. Implicit analysis of patient performance was presented with respect to STS activity. Acquired data during the game sessions refer to the players’ practice, and can be used to deliver both a summary and a deep description of their activity to clinical staff.

The use of ReMoVES for the care of frail elderly persons responds to a concrete need addressing an issue of both scientific and social interest. Bringing new technologies to the domestic level for frail elderly individuals is a turning point for their care, which benefits both the patients themselves, their families who are looking after them, and medical specialists in their practice.

### 6.1. Theoretical Contributions

The present study shows how simple solutions can deliver relevant information in the context of frail elderly persons’ care. On the one hand, the possibility to avoid wearable devices in favor of sensors such as depth cameras allow for expanding the use of such a system even without the need for therapists or caregivers to attend to patients. On the other hand, measurement accuracy is satisfactory for remote monitoring, as pathological movement can be detected from the data, and sessions can be analyzed and discussed as in Section 5.2.

The most important theoretical contribution is the definition of indicators that drive feature-extraction and data-processing operations in future studies. The reliability of data and concordance with respect to the literature enable the development of data-analysis and artificial-intelligence techniques for supporting clinical practice. For instance, due to the sequential nature of rehabilitation data collected by ReMoVES, long short-term memory (LSTM) [64] recurrent neural networks (RNNs) [65] can be used for data analysis.

### 6.2. Managerial Implications

The present study highlights the urgent need for officially recognizing IoT/ICT solutions for telerehabilitation and telemedicine in general. Even though acceptability towards such solutions is continuously increasing, it is still rare that such technologies are guaranteed and covered by national health services. For instance, in several Italian regions, the local health service has recently been operating in such a direction given the large number of elderly persons living there, recognizing the benefit that they may have from technologies such as the one described here. This is the case of Liguria where, to encourage better care of the elderly in the community, there are plans with funds for families to care for disabled relatives at home (in the context of the Silver Economy) [66].

In addition, necessary strategic plans for bridging the gap between the development of novel technologies and their extensive use by the population cannot disregard close contact with private companies.

To sum up, institutions should promote collaboration between universities and research centers in designing and prototyping novel solutions, high-tech companies that can convert a research product into a commercial one, and healthcare facilities as both experimental sites and final users to facilitate the spread of such technologies and consequent benefits to people.

### 6.3. Practical and Social Implications

There are three main contributions for what concerns practical and social implications of the present work.

First, from a clinical point of view, patients benefit from the use of ReMoVES in terms of help for dehospitalization, continuity of care, the personalization of plans of care, and engagement in activities.

Second, from an operative point of view, telerehabilitation helps clinical staff to also follow several patients when they cannot physically attend to them. This is very important in the time of the pandemic emergency, as it, for instance, allows for reducing time for moving from one patient’s house to the next. As a practical example, the Liguria region has relevant problems in terms of urban traffic; hence, home-based rehabilitation often causes therapists to lose time in traffic, augmenting work stress and eventually affecting the quality of the imparted treatment.

In the end, there were interesting social interactions among participants while practicing ReMoVES activities in the Centro di Riabilitazione facility. Patients enjoyed the experience both when playing themselves and when watching others carrying out the activities, thus promoting social inclusion. This may be the context of a stimulating future social experiment aimed at evaluating whether such a friendly atmosphere induced by ReMoVES activities can bring some sort of unexpected benefit.

### 6.4. Limitations and Future Research

The main limitation of the present study is the difficulty in performing a systematic comparison with similar studies, as mentioned in Section 3.

Concerning future developments, the next objective is the improvement of technical ReMoVES potential, for instance, by deploying the new Azure Kinect sensor, developing novel exergames, and including biometric sensors.

At the same time, the spectrum of diseases of which treatment also involves ReMoVES will be expanded. In particular, ReMoVES was recently awarded the Innovazione Digitale nella Sclerosi Multipla (Digital Innovation in Multiple Sclerosis) award, sponsored by Merck. As a result, a study on multiple sclerosis is being conducted.

To conclude, the emotional implication of the system in the population of patients will be investigated by means of specific questionnaires and the systematic analysis of patient feedback.

## Figures and Tables

**Figure 1 sensors-21-02719-f001:**
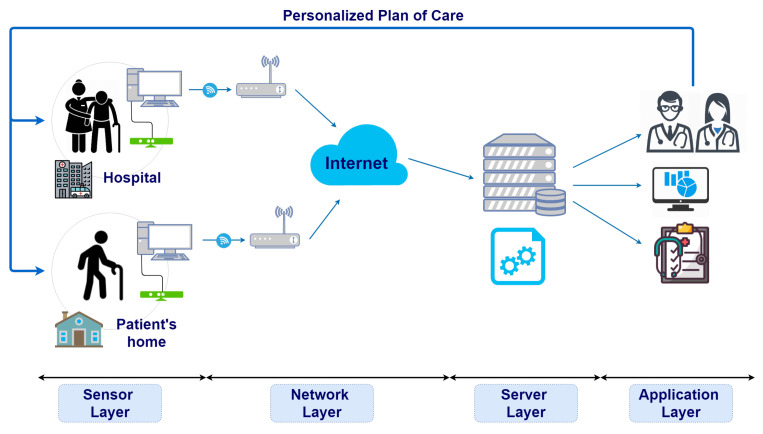
Architecture of remote-monitoring validation engineering system (ReMoVES). Each layer is depicted in the corresponding position.

**Figure 2 sensors-21-02719-f002:**
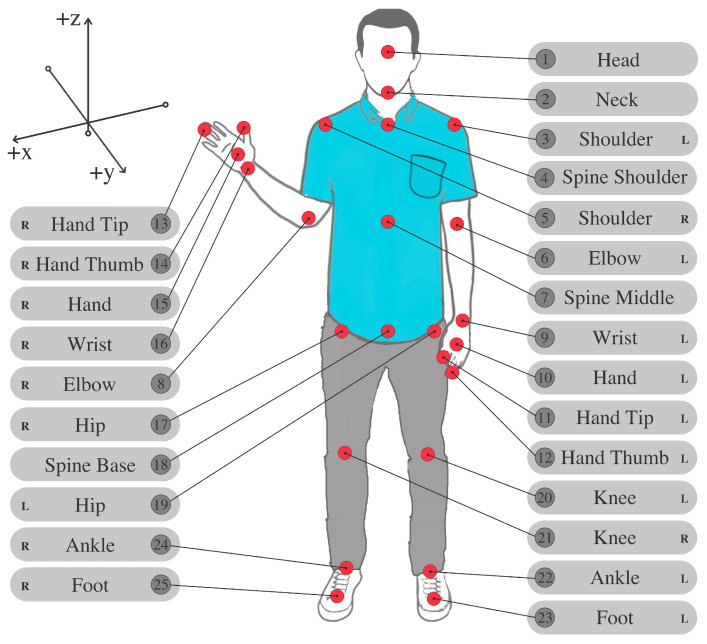
Skeleton-joint locations and names as captured by Microsoft Kinect sensor. Skeleton composed of 3D coordinates for each of its 25 joints.

**Figure 3 sensors-21-02719-f003:**
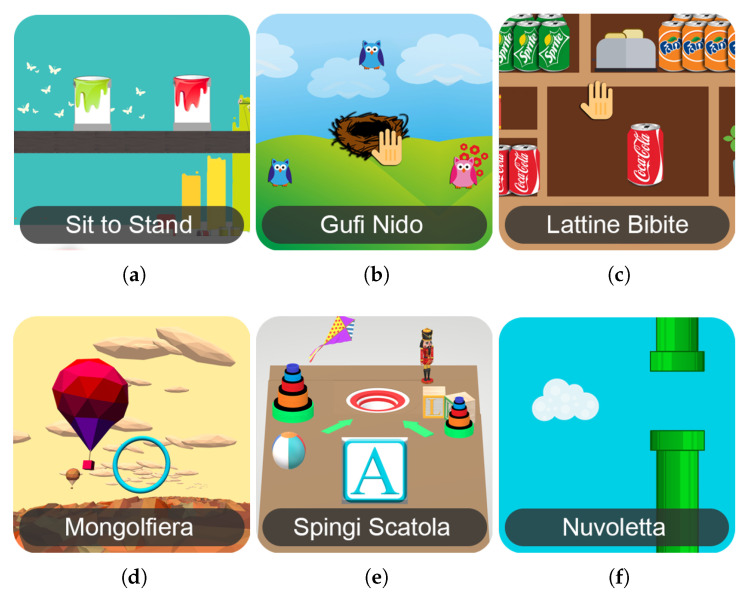
Thumbnails of Kinect exergames in current ReMoVES catalog. (**a**) Equilibrium Paint; (**b**) Owl Nest; (**c**) Shelf Cans; (**d**) Hot Air; (**e**) Push Box; (**f**) Flappy Cloud.

**Figure 4 sensors-21-02719-f004:**
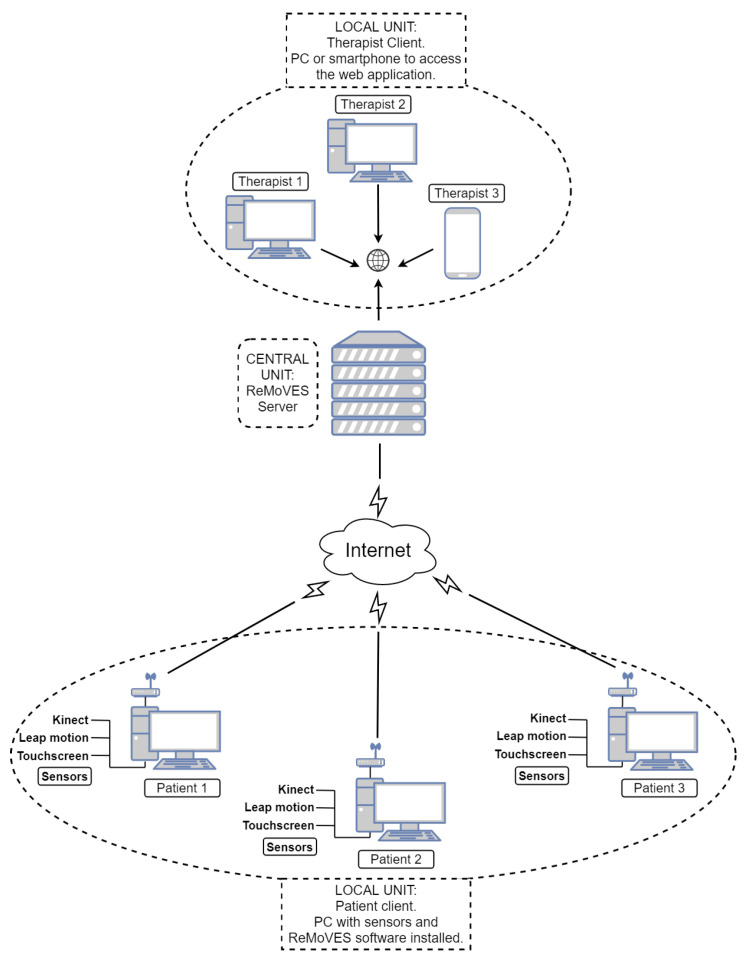
Hardware architecture of ReMoVES.

**Figure 5 sensors-21-02719-f005:**
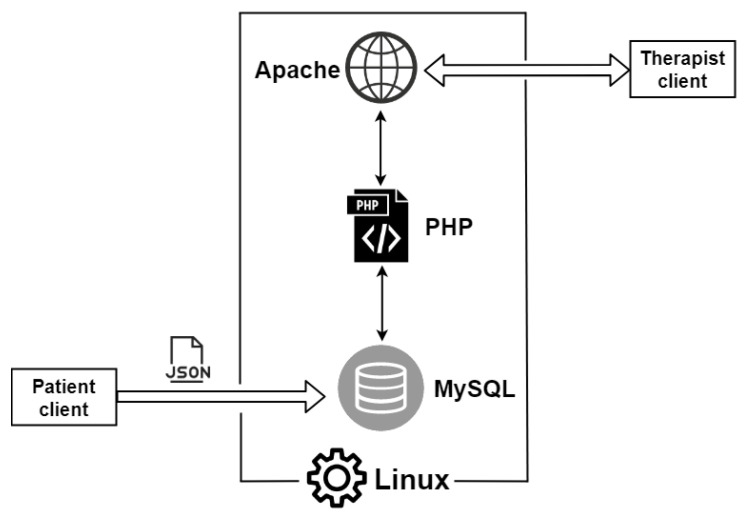
Software architecture of ReMoVES.

**Figure 6 sensors-21-02719-f006:**
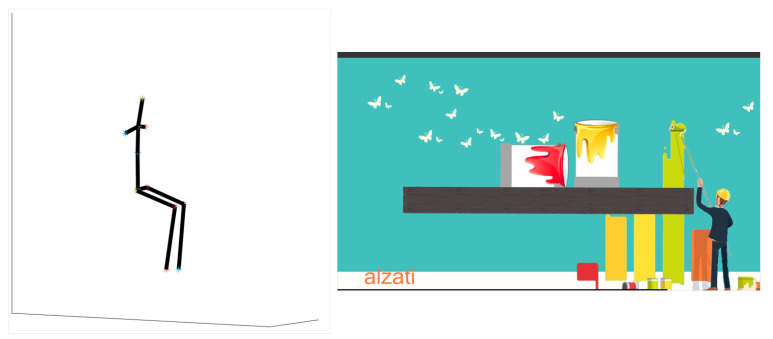
(**left**) Still reconstruction of movement for joints involved in indicator calculations (joints 1, 3–5, and 17–23 in Figure 2). Whole movement visible in the attached Appendix A. (**right**) Screenshot of Equilibrium Paint.

**Figure 7 sensors-21-02719-f007:**
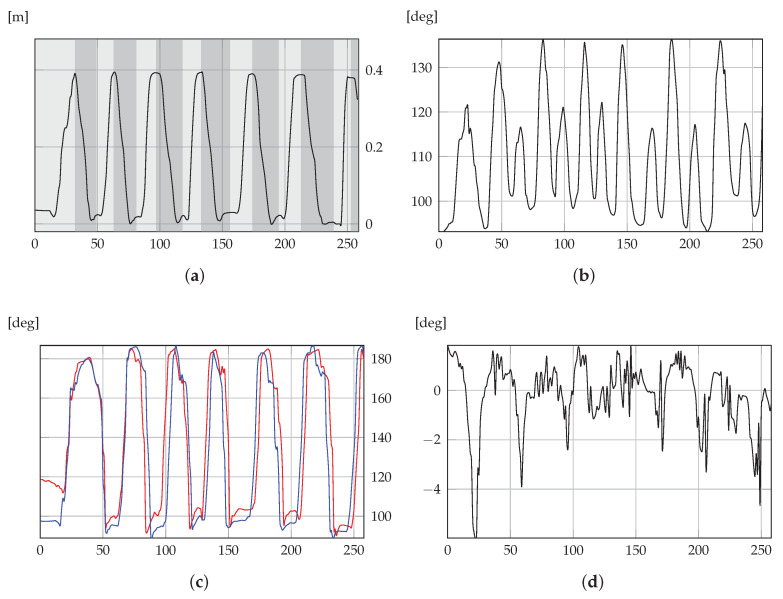
Graphs from Equilibrium Paint exergame. These graphic representations are available for clinical staff, so that deeper analysis is enabled throughout the whole session. (**a**) COM; (**b**) UBFA; (**c**) LLFA; (**d**) UBTA.

**Figure 8 sensors-21-02719-f008:**
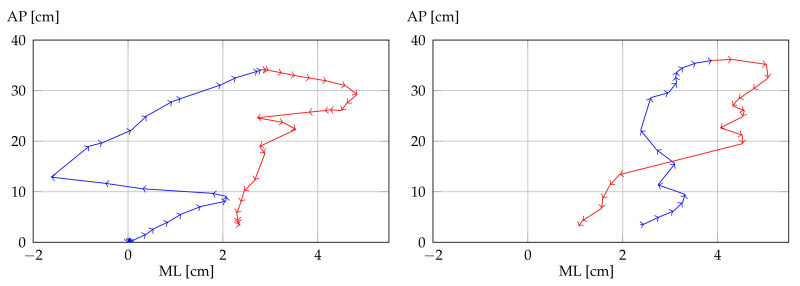
Representations of anteroposterior (AP) and mediolateral (ML) movements of center of mass in two consecutive ascending and descending phases. Blue lines, ascending phases; red lines, descending phases.

**Table 1 sensors-21-02719-t001:** Mean feature values. NSU, number of sit-up occurrences; UBFA, upper-body flexion angle; LLFA, lower-limb flexion angle; COM, center of mass; AP, anteroposterior; ML, mediolateral.

Feature	Mean Value
NSU	4.5±1.5
Stand UBFA range (deg)	79.92±6.71
Sit UBFA range (deg)	79.35±8.15
Stand LLFA (deg)	131.16±17.28
Sit LLFA (deg)	134.31±16.94
Stand UBTA (deg)	−0.67±1.91
Sit UBTA (deg)	−0.59±1.91
COM stand AP (cm)	0.36±0.09
COM sit AP (cm)	0.52±0.61
COM stand ML (cm)	0.08±0.02
COM sit ML (cm)	0.07±0.03
$UfV_{up}$ (m/s)	0.12±0.06
$UfV_{down}$ (m/s)	0.07±0.02

## Data Availability

Data described in the present study are released with this publication. They are available for download on GitLab (https://gitlab.com/NumIP/removes-fe-data/ accessed on 1 April 2021). The data are licensed under the Creative Commons Attribution 4.0 International license (CC BY-NC-SA 4.0).

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
