# Peer review of "A Solution for the Remote Care of Frail Elderly Individuals via Exergames"

_sensors, 2021, doi:10.3390/s21082719_

Round 1

Reviewer 1 Report

The paper is generally well written and focuses on an important topic to the academic and practice discourses. The following revisions are needed for the paper to be considered for publication.

1) The literature background needs some consolidation. The following could be a starting point:

Kankanamge, N., Yigitcanlar, T., Goonetilleke, A., & Kamruzzaman, M., (2020). How can gamification be incorporated into disaster emergency planning? A systematic review of the literature. International Journal of Disaster Resilience in the Built Environment, 11(4), 481-506, https://doi.org/10.1108/IJDRBE-08-2019-0054.

2) The methodological approach is in general sound and results are presented fine. However, right after '2. Materials and Methods' just add a paragraph to briefly reveal the methodological approach and the experiment (before going into details in the following sections. 

3) The only main issue is the limited discussion presented at the end of the paper (Discussion and Conclusion sections).  An engaging discussion on how the study findings will contribute to the academic knowledge, policymaking and practice is needed to address the 'so-what?' question.

4) Additionally, a careful final language editing is needed to polish the paper's language and flow of ideas.

Good luck at the revision. 

Author Response

We would like to thank the reviewers for their valuable comments. Here, we reply to their concerns and underline the modifications we performed, according to their suggestions. The novel or modified parts in the manuscript are written in red, in order to help reviewers for further readings.

The major issues were related to the introduction and literature review, and to the discussion.

Paper research gap and originality are now better presented at the end of introduction section, according to Reviewer3’s suggestions.

The literature background has been consolidated, starting from suggestions by Reviewer1 and Reviewer2. The paper now presents Section 2 for delimiting the clinical motivation of the present study, and Section 3 for covering the latest publications in the field.

As for the discussion, the structure suggested by Reviewer3 has been used, and that allowed also for addressing the concerns that Reviewer1 and Reviewer2 reported. Now, the discussion includes 4 subsections, i.e., Theoretical contributions - Managerial implications – Practical/social implications - Limitations and future research.

In addition, we would like to inform that the paper was edited by the MDPI English Editing Services for polishing the paper's language and flow of ideas. The title was changed according to the suggestion received.

A punctual response to reviewers’ comments follows.

Reviewer 1

1) The literature background needs some consolidation. The following could be a starting point:

Kankanamge, N., Yigitcanlar, T., Goonetilleke, A., & Kamruzzaman, M., (2020). How can gamification be incorporated into disaster emergency planning? A systematic review of the literature. International Journal of Disaster Resilience in the Built Environment, 11(4), 481- 506, https://doi.org/10.1108/IJDRBE-08-2019-0054.

Literature review has been improved: the suggested citation has been added (lines 82-83) and the whole Section 3 presents existing solutions in the framework and highlights both points in common and differences.

2) The methodological approach is in general sound and results are presented fine. However, right after '2. Materials and Methods' just add a paragraph to briefly reveal the methodological approach and the experiment (before going into details in the following sections.

Lines 222-230 are for the added paragraph to briefly reveal the methodological approach and the experiments.

3) The only main issue is the limited discussion presented at the end of the paper (Discussion and Conclusion sections). An engaging discussion on how the study findings will contribute to the academic knowledge, policymaking and practice is needed to address the 'so- what?' question.

Lines 556-644 are for the improved discussion section. It is now organized to highlight contributions from different points of view:

Theoretical contributions: academic knowledge contribution.

Managerial implications: policymaking.

Practical/social implications: practice.

Limitations and future research.

Conversely, the analysis of the presented graphs was moved in the section for experimental results.

Reviewer 2 Report

The work proposes ReMoVES platform that caters to the specific needs of frail elders. The architecture is designed for use at both rehabilitation and at patients homes. The system is user-friendly and comfortably usable by persons who are not familiar with technology. The platform ReMoVES is currently used in five centers among hospitals, clinics, and facilities, involving more than 200 patients and resulting in more than 2000 rehabilitation sessions. The contributions are well delimited which is the use of the IoT system in the context of frail elders care. The ease-of-use of the presented solution allows the frail elders to autonomously access the services provided which is not requiring controllers and avoiding wearable sensors for data collection.

The topic fits very well the scope of the journal. The structure of the paper is clear, the language is proper and contributions are well delimited.

In the end of introduction section needs a paragraph describing the manuscript organising. 

Authors should indicate more recent work in this field such as doi.org/10.3390/info11090438 and among others. Please, search in the literature in order to update the related work section. The ReMoVES is very relevant, however it is important to cover the latest publications in the field. 

In other to better clarify the architecture, should be provided a picture of hardware and software architecture to describe the components and relationship between layers.  

Authors should explain why the real world case study is involved *only* thirteen frail elders (6 females and 7 males). 

The discussion section is very limited! So, the importance of the ReMoVES framework deserv a lot of discussion. Authors must explore this section in order to better quality of the manuscript.

The manuscript needs a revision in order to correct some typos.

Author Response

We would like to thank the reviewers for their valuable comments. Here, we reply to their concerns and underline the modifications we performed, according to their suggestions. The novel or modified parts in the manuscript are written in red, in order to help reviewers for further readings.

The major issues were related to the introduction and literature review, and to the discussion.

Paper research gap and originality are now better presented at the end of introduction section, according to Reviewer3’s suggestions.

The literature background has been consolidated, starting from suggestions by Reviewer1 and Reviewer2. The paper now presents Section 2 for delimiting the clinical motivation of the present study, and Section 3 for covering the latest publications in the field.

As for the discussion, the structure suggested by Reviewer3 has been used, and that allowed also for addressing the concerns that Reviewer1 and Reviewer2 reported. Now, the discussion includes 4 subsections, i.e., Theoretical contributions - Managerial implications – Practical/social implications - Limitations and future research.

In addition, we would like to inform that the paper was edited by the MDPI English Editing Services for polishing the paper's language and flow of ideas. The title was changed according to the suggestion received.

A punctual response to reviewers’ comments follows.

Reviewer 2

In the end of introduction section needs a paragraph describing the manuscript organising.

A paragraph for describing the manuscript organizing was added.

Authors should indicate more recent work in this field such as doi.org/10.3390/info11090438 and among others. Please, search in the literature in order to update the related work section. The ReMoVES is very relevant, however it is important to cover the latest publications in the field.

Starting from the suggested paper, the literature review has been improved by mentioning more recent related works. The whole Section 3 presents existing solutions in the framework and highlights both points in common and differences.

In other to better clarify the architecture, should be provided a picture of hardware and software architecture to describe the components and relationship between layers.

Figure 4 and 5 and the description are for better clarifying the architecture.

Authors should explain why the real world case study is involved *only* thirteen frail elders (6 females and 7 males).

The present study is at a preliminary feasibility stage, aimed at evaluating the use of ReMoVES in a real-world scenario. For such a reason, the experimental study involved 13 frail elders who took part several times to the rehabilitation sessions via ReMoVES. The feedback from the present work will drive further applications involving more patient.

The discussion section is very limited! So, the importance of the ReMoVES framework deserves a lot of discussion. Authors must explore this section in order to better quality of the manuscript.

The improved discussion section was added. In this revised version we highlighted the contributions and implications from the following points of view: Theoretical contributions, Managerial implications, Practical/social implications.

Conversely, the analysis of the presented graphs was moved in the section for experimental results.

Reviewer 3 Report

I am pleased to have the opportunity to review this research paper. This study attempted to explore a solution for remote caring of frail elderly via exergames. Although the topic of this research study is interesting and fits within the journal scope, I think authors should apply the comments indicated below to increase the quality of research justification, contributions and findings.

First of all, paper research gap. Please improve this part in introduction section: This part is very general and lacked alignment to the research findings, no discussion was provided to derive the implication from. Theoretical and pragmatics implication are vague and need to be better aligned with this paper theoretical underpinnings and proposed process. Furthermore, there is insufficient support and weak arguments in support of the objective that is proposed as well as the model developed. In the final part of introduction, the manuscript structure should be summarised as well as the objetives proposed, originality and gap that would be covered. Also how the author will perform the methodology.

What is the originality of this research?  Paper research gap and originality should be better presented at the end of introduction section. Please use this paper and make a citation to solve this task as the author of this study referred to the use of UGC and online reviews: doi: https://doi.org/10.1016/j.jik.2020.08.001

Please consider this structure for manuscript final part.

Conclusion

Managerial Implication

Practical/Social Implications

Limitations and future research

Discussion needs to be a coherent and cohesive set of arguments that take us beyond this study in particular, and help us see the relevance of what authors have proposed.  Author need to contextualise the findings in the literature, and need to be explicit about the added value of your study towards that literature. Also other studies should be cited to increase the theoretical background of each of the method used. Findings should be contextualised in the literature and should be explicit about the added value of the study towards the literature. Please use this citation to copy the style and make a citation:  https://doi.org/10.1016/j.ijinfomgt.2021.102331

Questions to be answered:

What practical/professional and academic consequences will this study have for the future of scientific literature (theoretical contributions)?

Why is this study necessary? Again, the authors should make clear arguments to explain what is the originality and value of the proposed model to explore UGC reviews. This should be stated in the final paragraphs of introduction and conclusion sections.

Author Response

We would like to thank the reviewers for their valuable comments. Here, we reply to their concerns and underline the modifications we performed, according to their suggestions. The novel or modified parts in the manuscript are written in red, in order to help reviewers for further readings.

The major issues were related to the introduction and literature review, and to the discussion.

Paper research gap and originality are now better presented at the end of introduction section, according to Reviewer3’s suggestions.

The literature background has been consolidated, starting from suggestions by Reviewer1 and Reviewer2. The paper now presents Section 2 for delimiting the clinical motivation of the present study, and Section 3 for covering the latest publications in the field.

As for the discussion, the structure suggested by Reviewer3 has been used, and that allowed also for addressing the concerns that Reviewer1 and Reviewer2 reported. Now, the discussion includes 4 subsections, i.e., Theoretical contributions - Managerial implications – Practical/social implications - Limitations and future research.

In addition, we would like to inform that the paper was edited by the MDPI English Editing Services for polishing the paper's language and flow of ideas. The title was changed according to the suggestion received.

A punctual response to reviewers’ comments follows.

Reviewer 3

First of all, paper research gap. Please improve this part in introduction section: This part is very general and lacked alignment to the research findings, no discussion was provided to derive the implication from. Theoretical and pragmatics implication are vague and need to be better aligned with this paper theoretical underpinnings and proposed process. Furthermore, there is insufficient support and weak arguments in support of the objective that is proposed as well as the model developed. In the final part of introduction, the manuscript structure should be summarised as well as the objetives proposed, originality and gap that would be covered. Also how the author will perform the methodology.

What is the originality of this research? Paper research gap and originality should be better presented at the end of introduction section. Please use this paper and make a citation to solve this task as the author of this study referred to the use of UGC and online reviews: doi: https://doi.org/10.1016/j.jik.2020.08.001

Introduction and literature review has been improved.

According to the suggested citation, the areas of research on the use of Data Science in Digital Marketing, where the present work finds applications, have been delimited.

Paper research gap and originality are now better presented at the end of introduction.

Section 3 presents existing solutions in the framework and highlights both points in common and differences, and therefore why the present study was necessary.

Please consider this structure for manuscript final part. Conclusion Managerial Implication Practical/Social Implications Limitations and future research

Discussion needs to be a coherent and cohesive set of arguments that take us beyond this study in particular, and help us see the relevance of what authors have proposed. Author need to contextualise the findings in the literature, and need to be explicit about the added value of your study towards that literature. Also other studies should be cited to increase the theoretical background of each of the method used. Findings should be contextualised in the literature and should be explicit about the added value of the study towards the literature. Please use this citation to copy the style and make a citation: https://doi.org/10.1016/j.ijinfomgt.2021.102331

Questions to be answered:

What practical/professional and academic consequences will this study have for the future of scientific literature (theoretical contributions)?

Why is this study necessary? Again, the authors should make clear arguments to explain what is the originality and value of the proposed model to explore UGC reviews. This should be stated in the final paragraphs of introduction and conclusion sections.

The improved discussion section was added. In this revised version we tried to discuss the work beyond the present application (indeed, the analysis of the presented graphs was moved in the section for experimental results).

Following the suggested citation, the discussion is now organized to highlight contributions from different points of view:

Theoretical contributions: academic knowledge contribution.

Managerial implications: policymaking.

Practical/social implications: practice.

Limitations and future research.

Round 2

Reviewer 1 Report

adequate improvements were made In the revised paper, it and now be accepted for publication.

Reviewer 3 Report

The authors have addressed correctly the indicated changes. This reviewer has no additional comments. Congratulations to the authors for the development of this manuscript.